# Exploring Candidate Gene Studies and Alexithymia: A Systematic Review

**DOI:** 10.3390/genes15081025

**Published:** 2024-08-04

**Authors:** Yazmín Hernández-Díaz, Alma Delia Genis-Mendoza, Thelma Beatriz González-Castro, Ana Fresán, Carlos Alfonso Tovilla-Zárate, María Lilia López-Narváez, Isela Esther Juárez-Rojop, Humberto Nicolini

**Affiliations:** 1División Académica Multidisciplinaria de Jalpa de Méndez, Universidad Juárez Autónoma de Tabasco, Jalpa de Méndez 86205, Mexico; yazmin.hdez.diaz@gmail.com; 2Hospital Psiquiátrico Infantil “Juan N. Navarro”, Tlalpan 14080, Mexico; adgenis@inmegen.gob.mx; 3Subdirección de Investigaciones Clínicas, Instituto Nacional de Psiquiatría Ramón de la Fuente Muñiz, Ciudad de México 14370, Mexico; a_fresan@yahoo.com.mx; 4División Académica Multidisciplinaria de Comalcalco, Universidad Juárez Autónoma de Tabasco, Comalcalco 86650, Mexico; alfonso_tovillaz@yahoo.com.mx (C.A.T.-Z.); dralilialonar@yahoo.com.mx (M.L.L.-N.); 5División Académica de Ciencias de la Salud, Universidad Juárez Autónoma de Tabasco, Jalpa de Méndez 86205, Mexico; iselajuarezrojop@hotmail.com; 6Laboratorio de Genómica de Enfermedades Psiquiátricas y Neurodegenerativas, Instituto Nacional de Medicina Genómica, Ciudad de México 14610, Mexico

**Keywords:** alexithymia, genes, biomarkers

## Abstract

Background: Alexithymia is a trait involving difficulties in processing emotions. Genetic association studies have investigated candidate genes involved in alexithymia’s pathogenesis. Therefore, the aim of the present study was to perform a systematic review of the genetic background associated with alexithymia. Methods: A systematic review of genetic studies of people with alexithymia was conducted. Electronic databases including PubMed, Scopus, and Web of Science were searched for the study purpose. We used the words “Alexithymia”, “gene”, “genetics”, “variants”, and “biomarkers”. The present systematic review was performed following the Preferred Reporting Items for Systematic reviews and Meta-Analyses statement. We found only candidate gene studies. A total of seventeen studies met the eligibility criteria, which comprised 22,361 individuals. The candidate genes associated with alexithymia were the serotoninergic pathway genes solute carrier family 6 member 4 (*SLC6A4*), serotonin 1A receptor (*HTR1A*), and serotonin 1A receptor (*HTR2A*); the neurotransmitter metabolism genes dopamine receptor D2 (*DRD2*), ankyrin repeat and kinase domain containing 1 (*ANKK1*), catechol-o-methyltransferase (*COMT*), brain-derived neurotrophic factor (*BDNF*), and oxytocin receptor (*OXTR*); and other pathway genes, vitamin D-binding protein (*VDBP*), tumor protein P53 regulated apoptosis inducing protein 1 (*TP53AIP1*), Rho GTPase Activating Protein 32 (*ARHGAP32*), and transmembrane protein 88B (*TMEM88B*). Conclusion: The results of this study showed that only case–control gene studies have been performed in alexithymia. On the basis of our findings, the majority of alexithymia genes and polymorphisms in this study belong to the serotoninergic pathway and neurotransmitter metabolism genes. These data suggest a role of serotoninergic neurotransmission in alexithymia. Nevertheless, more and future research is required to learn about the role of these genes in alexithymia.

## 1. Introduction

Concerning the epidemiological data, it is difficult to estimate the prevalence of alexithymia due to there being no clear diagnostic criteria. In the general population, around 10% of people have problematically high levels of alexithymia [1]. Nevertheless, in patients with physical illnesses, the prevalence of alexithymia could reach more than 60% [2]. Moreover, gender differences have been pointed out; for example, large-scale studies in general population samples reported that males showed higher scores than females [3,4]. In fact, it has been reported that the prevalence of alexithymia ranges from 9 to 17% for men and 5 to 10% for women [5]. Actually, one of the most alarming facts about alexithymia is that the prevalence increases in adolescents and younger people [6,7]. Specifically, in females, pubertal maturation is associated with difficulties in identifying and describing feelings [8]. Hence, developmental changes in alexithymia should be considered during adolescence. In this sense, generally, the development of alexithymia is considered to begin in childhood, and in adult life, it comes to be a stable psychiatric trait [9,10]. Alexithymia is a personality trait characterized by an impaired ability to be aware of and describe one’s feelings; in fact, the term literally means “no words for feeling” [11]. Specifically, persons with alexithymia manifest some cognitive and affective features. For example, they have difficulty in finding words to identify and describe feelings, emotions, or any physical sensations to others [11].

Many adverse health issues are associated with alexithymia, among them depression, anxiety, and various somatic comorbidities [12,13]. Given the fact that alexithymia is highly correlated with these psychopathologies, it is plausible that alexithymia could be a lifetime risky health behavior [14]. Therefore, some studies are searching for the predisposing factors for the development of this lack of emotional awareness [15,16]. Recently, studies have suggested that a genetic component could be linked to alexithymia [17,18]. In particular, a twin study indicated that familial influences contribute to alexithymia. Specifically, this study supported the idea that genetic factors are implicated in the externally oriented thinking domain [19]. Dysfunction in the dopaminergic and serotoninergic systems has been shown to have a prominent genetic impact on the pathogenesis of alexithymia. However, at the moment, there have not been any previous relevant systematic reviews that show the candidate genes that have been studied in alexithymia. Therefore, the objective of this systematic review was to describe the candidate genes and polymorphisms associated with alexithymia to provide an understanding of the role of these genes in alexithymia.

## 2. Materials and Methods

### 2.1. Design

The guidelines of the Preferred Reporting Items for Systematic Review and Meta-Analysis (PRISMA) were followed for the methodology and served as a template for the present review; Appendix A. 

### 2.2. Search Strategy

The studies were identified in PubMed, Scopus, and Web of Science without date restriction. The first and corresponding authors generated the following search string: (“Alexithymia”) and (“gene” OR “genetics” OR “biomarkers” OR “variants”). To supplement these database searches, the references in the identified articles and reviews were hand-searched for relevant studies, and this resulted in one article that was not captured by the search strategy. The database search was started in August 2023, then updated until March 2024.

#### 2.2.1. Eligibility Criteria

##### Type of Studies

According to our aim, the articles included in the present work met the following criteria: (a) observational (case–control and cross-sectional), (b) published in a peer-reviewed journal and indexed in journal citation reports, and (c) published in English; further, (d) if the theoretical model used for describing the alexithymia construct is based on the Toronto Alexithymia Scale-20 (TAS-20), then we include the TAS-20 for assessing alexithymia. 

##### Exclusion Criteria

The exclusion criteria were as follows: reviews, editorials, opinion papers, and other studies presenting non-original data; conference papers; and articles providing incomplete data or insufficient genotype frequency information.

### 2.3. Evaluation of Methodological Quality

The quality assessment was performed by the same reviewers from the search strategy. Eligibility criteria and data extraction were realized using the Newcastle–Ottawa Scale (NOS). The NOS includes three sections: selection (four points), comparability (two points), and exposure (three points). The included studies were categorized as low (from 0 to 3 points), moderate (from 4 to 6 points), or high quality (from 7 to 9 points); see Appendix A.

### 2.4. Data Extraction and Analysis

The data extracted from the cases and controls of the selected studies were the clinical status, sample size, age, gender, and distribution of genotypes and alleles. Furthermore, we extracted some general data on the eligible articles: the authors’ names, year of publication, patient ethnicity, country, diagnostic criteria, biological samples, and lab methods. All the data were extracted independently by the first and corresponding authors according to the set retrieval strategy and the selection criteria. Additionally, in disagreements, a third author was consulted in order to reach an agreement.

## 3. Results

### 3.1. Characteristics of the Studies

Figure 1 shows a flow diagram of the complete process selection. Seventeen studies met the selection criteria previously mentioned. The included populations were mainly distributed in Europe [17,20,21,22,23,24,25,26,27] and Asia [18,28,29,30,31,32,33]; only one study population was from America [34]. The total number of individuals included in this systematic review was 22,361. The sample size in the studies ranged from 80 to 6267. According to the NOS, the quality scores of the included studies ranged from 6 to 9.

From these studies, the candidate genes associated with alexithymia risk were solute carrier family 6 member 4 (*SLC6A4*), serotonin 1A receptor (*HTR1A*), serotonin 1A receptor (*HTR2A*), dopamine receptor D2 (*DRD2*), ankyrin repeat and kinase domain containing 1 (*ANKK1*), catechol-o-methyltransferase (*COMT*), brain-derived neurotrophic factor (*BDNF*), oxytocin receptor (*OXTR*), vitamin D-binding protein (*VDBP*), tumor protein P53 regulated apoptosis inducing protein 1 (*TP53AIP1*), Rho GTPase Activating Protein 32 (*ARHGAP32*) and transmembrane protein 88B (*TMEM88B*). The variants significantly associated (*p* value of <0.05) with alexithymia were described in these works and are shown in Table 1. For a better understanding, we grouped the results of the studies according to the biological pathway involved: the serotoninergic pathway, neurotransmitter metabolism, and other pathways.

### 3.2. Genes in the Serotoninergic Pathway

SLC6A4: The serotonin transporter (SERT), also named 5-Hydroxy Tryptamine Transporter (5-HTT), is encoded in the *SLC6A4* gene [35]. The human *SCL6A4* gene is composed of 14 exons spanning approximately 40 Kb and located on chromosome 17q11.2 [36]. The reported evidence indicates a strong effect of this gene on mood behavior disorders [35]. Concerning alexithymia, there have been studies searching for the role of *SLC6A4* gene variants. The promoter region of the *SLC6A4* harbors a highly polymorphic region, named 5-HTTLPR [37]. In a report of 304 Japanese subjects, those with the L/L genotype of HTTLPR showed significantly higher TAS-20 scores than did those with the L/S or S/S genotype [28]. In agreement, in an Italian Caucasian population, alexithymia items were analyzed according to gender. The study identified that L/L in males was positively correlated with the DIF (difficulty identifying feeling) domain [24]. This supports the assertion that in bi- or tri-allelic 5-HTTLPR polymorphism, the L-allele shows a statistical association of low expression with an increased TAS-20 score [25].

HTR1A and HTR2A: The reported evidence indicates a direct impact of specific serotonin receptor modulation on behaviors [38]. Hence, there are studies that have focused on their effects on alexithymia [29]. The *HTR1A* and *HTR2A* genes are located on chromosome 5q12.3, with a size of ≈4 KB, and chromosome 13q14.2, with a size of ≈66 KB, respectively [38]. One of the most studied variants of the *HTR2A* gene is rs6311; from this, it was reported that the G allele is related to a higher TAS-20 score and with mental health problems mediated by alexithymia [29]. Moreover, rs6295 of *HTR1A* and rs6311 of *HTR2A* were analyzed as risk factors for alexithymia. Specifically, a study evaluated whether these factors show an effect with childhood trauma on patients with alexithymia. They found that childhood trauma is associated with rs6295 (*HTR1A*) interactions in males; in more detail, the presence of the G allele is linked to an increase in TAS-20 scores [26]. In addition, rs6295 *HTR1A* gene polymorphism has been studied as to whether it modulates individuals’ alexithymic characteristics and attachment orientation. Interestingly, it was reported that CG-GG genotype carriers seem to be less comfortable in close relationships and show significantly higher TAS-20 scores [31]. Furthermore, the relationship of alexithymia in depressed patients with chronic hepatitis C has been studied, and patients with higher levels of alexithymia and the *HTR1A*-G allele are more predisposed to experiencing IF-induced depression [20].

### 3.3. Genes in Neurotransmitter Metabolism

DRD2 and ANKK1: DRD2 is a G-protein-coupled receptor that inhibits adenylyl cyclase activity. The *DRD2* gene is located in the long arm of chromosome 11 (band 23.2) [39]. The cytogenic band of the *ANKK1* gene is 11q23.3 and, therefore, is closely related to the *DRD2* gene [40]. The ANKK1 protein is a member of a kinase superfamily involved in signal transduction pathways. Hence, this protein is strongly related to the antipsychotic pathway [41]. Mutations in the *DRD2/ANKK1* genes have been related to several psychiatric diagnoses [42,43]. In particular, the A1+ allele of the DRD2/ANKK1 TaqI A SNP (rs1800497) is related to alexithymia, and prior emotional abuse leads to a higher risk of alcohol problems [34]. Further evidence that supports this relationship was reported in a study of 664 healthy Germans, which observed that subjects with the DRD2/ANKK1 A1 allele had higher scores in the DIF subscale of TAS-20 [23].

COMT: The *COMT* gene is located on chromosome 22q11.21 and is composed of approximately 28 thousand bases [44]. COMT participates in the transfer of a methyl group from S-adenosylmethionine to catecholamines (e.g., dopamine, epinephrine, and norepinephrine). Therefore, there are several lines of evidence indicating a strong association with mood disorders [45]. The most commonly studied variant of the *COMT* gene is Val108/158Met polymorphism, which is frequently proposed as a biomarker of risk. In particular, it has been seen that Val/Val carriers have significantly higher TAS-20 scores than do Met/Met carriers [33]. Furthermore, the association between alexithymia and the Val108/158Met SNP has been linked to other factors. For example, a study with 244 patients with obsessive–compulsive disorder (OCD) found that the Val/Val genotype is associated with significantly higher DDF subdimension scores [32]. The correlation between alexithymia and hypervigilance to pain has also been evaluated. A study reported a significant positive correlation between the DDF subdimension and attention to changes in pain in Met carriers but not in Val/Val individuals [18].

BDNF: The *BDNF* gene, located on 11p14.1, is a member of the nerve growth factor family of proteins [46,47]. Due to its participation in maintaining brain function and its multiple gene variants, this gene has been linked to the regulation of the stress response and the biology of mood disorders [47]. The Val66Met SNP (rs6265) is the most commonly proposed biomarker of the *BDNF* gene. In alexithymia, association studies have shown that, among healthy subjects, carriers of at least one 66Met allele have the highest scores in the total TAS-20 scale and its DIF subdimension [23]. Nevertheless, another report showed that BDNF Val66Met had no significant main effect on TAS-20 scores in a Caucasian population [21].

OXTR: The *OXTR* gene is situated on chromosome 3p25.3 and encompasses 28,360 bases [48]. Oxytocin receptor belongs to the G-protein-coupled receptor family. In humans, it is present in the limbic system, including the amygdala and hypothalamus [49]. One *OXTR* gene variant related to social behavioral phenotypes is rs53576. In 195 healthy individuals studied, an interaction of rs53576 and alexithymia was observed, with GG homozygosity being significantly associated with more severe alexithymia [22]. Nevertheless, in single marker and haplotype association analyses of *OXTR* gene variants (rs237885, rs237887, rs2268490, rs4686301, rs2254298, rs13316193, rs53576, and rs2268498), none showed a significant association in OCD patients [30].

### 3.4. Other Genetic Pathways

Low vitamin D serum levels may lead to lower expression of the rate-limiting enzyme in serotonin synthesis, which is linked to an increased presynaptic shortfall of available serotonin [19]. In a general population, a study genotyped the *VDBP* gene variants rs4588 and rs7041; nevertheless, no significant association with alexithymia was reported [17]. Additionally, another study analyzed exome functional variants in association with alexithymia. The exome association analysis detected four significant variants (*ABCB4* gene: rs45575636, *TP53AIP1* gene: rs35942033, *ARHGAP32* gene: rs35287114, and *TMEM88B* gene: rs144957058) associated with alexithymia. There is not much information about these genes and their biological pathways in the brain. ABCB4/Abcb4 mRNA expression is low but detectable in the human brain and rat choroid plexus; it is an extremely effective phosphatidylcholine transporter [50]. TP53AIP1 is thought to play an important role in mediating p53-dependent apoptosis [51]. Meanwhile, ARHGAP32 encodes a neuron-associated GTPase-activating protein and is involved in early brain development, including the extension of axons and dendrites and the postnatal remodeling and fine-tuning of neural circuits [52,53]. Little is known about the *TMEM88B* gene or its protein, so further studies are necessary to understand its functional characterization.

## 4. Discussion

In the present study, we identified candidate genes in association with alexithymia. Seventeen studies were included, with 22,361 subjects studied. Multiple factors have been suggested in the pathogenesis and development of alexithymia, for which several genetic biomarkers can be observed. From these, concerning alexithymia, the principal genes are linked to the serotoninergic system and neurotransmitter metabolism. We found that 12 genes and 13 polymorphisms have been associated with alexithymia, as shown in Figure 2.

The first possible link between alexithymia and the serotoninergic system is HTTLPR polymorphism. Evidence shows that the short variant (S-allele) has been linked to reduced transcriptional activity compared with the long variant (L-allele) [36]. Interestingly, in alexithymia association studies, a significant relationship with the presence of the L-allele was observed [28]. Nevertheless, we should take into consideration that essential ethnic differences in 5-HTTLPR frequency could be responsible for this inconsistency [54]. Other candidate genes from the serotoninergic system are the serotonin receptors, especially the rs6295 and rs6311polymorphisms of the *HTR1A* and *HTR2A* genes, respectively. If these polymorphisms proposed for the transporter (HTTLPR) and receptor (rs6295 and rs6311) genes lead to changes in 5-HT concentrations in the synaptic cleft or amygdala, alexithymic behavioral traits will subsequently appear. This is the principal reason why the serotonin pathway is important for drug development [55,56]. Furthermore, a modulatory effect has been reported for 5-HTTLPR polymorphism on the connectivity of the ventral attention network and impaired response inhibition, which are implicated in stimulus-driven attentional control [57]. Serotonin is responsible for regulating a wide variety of mood disorders, mostly through the action of serotonin receptors (HTR1A and HTR2A) and serotonin transporter (SERT) [55]. Changes in and disruption to the serotoninergic system are strongly linked to mood disorders. Therefore, the different contributions of 5HT receptors (HTR1A and HTR2A) in the brain should be considered in alexithymia [56]. Other important pathways related to behavior are those involved in neurotransmitter metabolism, such DRD2/ANKK1, COMT, BDNF, and oxytocin receptor. First, the A1+ allele of the DRD2/ANKK1 gene is statistically linked to alexithymia [21,23]. The A1 allele affects the DRD2 receptor density and increases l-DOPA activity; indeed, this allele is more strongly associated with negative emotions or cognitive alterations [58]. One explanation for this is that these receptors are primarily expressed in sub-cortical regions like the nucleus accumbens and caudate putamen, where they are involved in the modulation of emotions, highlighting their importance in alexithymia [59]. Concerning the Val108/158Met polymorphism of the *COMT* gene, the Val carriers are the individuals with a higher risk of alexithymia [32,33]. The Val form of the COMT enzyme is more active and leads to lower levels of synaptic dopamine in the prefrontal cortex [60]. Specifically, COMT Val108/158Met variation is linked to increased activation in prefrontal areas, causing reduced ability in the specific domain and increased cortico-limbic activity, allowing emotional dysregulation; these effects point to an association between the COMT genotype and susceptibility to affective disorders such as alexithymia [61]. Therefore, the Val108/158Met variant is highly related in genetic association studies with *ANKK1/DRD2* gene variants.

On the other hand, for the *BDNF* gene, the candidate variant was Val66Met, specifically the 66Met allele, in patients with alexithymia. In the BDNF protein, the substitution of Methionine disrupts the activity-dependent secretion of BDNF at the synapse, subsequently affecting hippocampal functions [62]. The above allows us to associate the *BDNF* gene with predisposition to specific personality traits, including fatigue, frustration, worrying, pessimism, shyness, and persistence, among others. Finally, for the *OXTR* gene, one candidate variant for alexithymia risk is the rs53576 polymorphism. This polymorphism is related to empathic responses. More precisely, G allele carriers show greater empathic responses than do AA homozygotes [63]. In particular, the expression of the previously mentioned genetic biomarkers of neurotransmitter metabolism is related to specific brain areas such as the amygdala or posterior cingulate cortex. Specifically, these brain areas are involved in some emotional processes (e.g., emotional stimuli, emotional experience, self-referential thinking) [64,65].

Alexithymia has been implicated in the prognosis and development of diverse diseases. Nevertheless, health workers seem to rarely consider alexithymia during the treatment of any disease. Therefore, we consider it very important to understand the complex genetic network leading to the onset and progression of alexithymia. Nevertheless, the findings present some limitations. First, there has been a limited number of genetic association studies on alexithymia in comparison with other psychiatric diseases or traits such as schizophrenia or depression. In particular, we did not observe any GWASs in alexithymia. However, this was an exploratory view of the genetic background of alexithymia candidate biomarkers. This opens the door for future GWASs, along with epigenetic and expression studies. Another point to observe is the limitations of the data available regarding the variable of gender. It has been established that gender could cause an imbalance because a combination of biological, psychological, and sociocultural factors can be involved in alexithymia. Furthermore, it is more appropriate to identify discrete variables such as gender in further studies.

## 5. Conclusions

The main candidate genes associated with alexithymia are from the serotoninergic pathway (*SLC6A4*, *HTR1A* and *HTR2A*) and neurotransmitter metabolism (*DRD2/ANKK1*, *COMT*, *BDNF*, and *OXTR*). Furthermore, there are other genes related to alexithymia risk, such as *VDBP*, *TP53AIP1*, *ARHGAP32*, and *TMEM88B*. We believe that these candidate genes will be useful additions to neuropsychological assessments of alexithymia. Understanding the genetic background of alexithymia in future studies could help in assessing the prognosis and response to a specific treatment.

## Figures and Tables

**Figure 1 genes-15-01025-f001:**
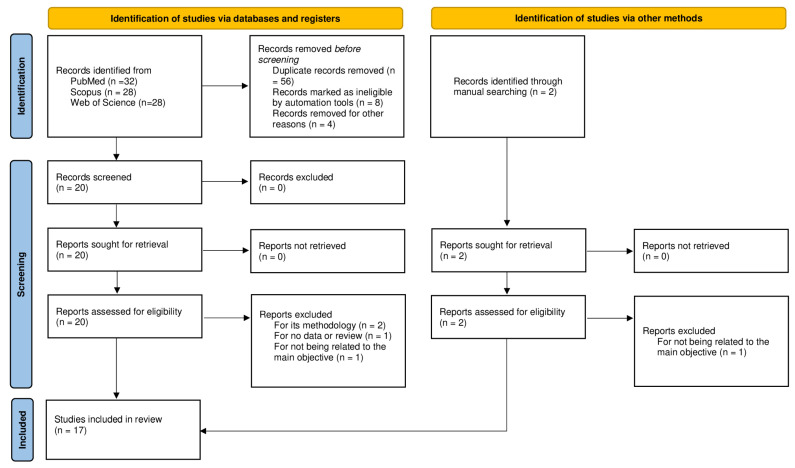
Flow diagram of the selection criteria for this systematic review.

**Figure 2 genes-15-01025-f002:**
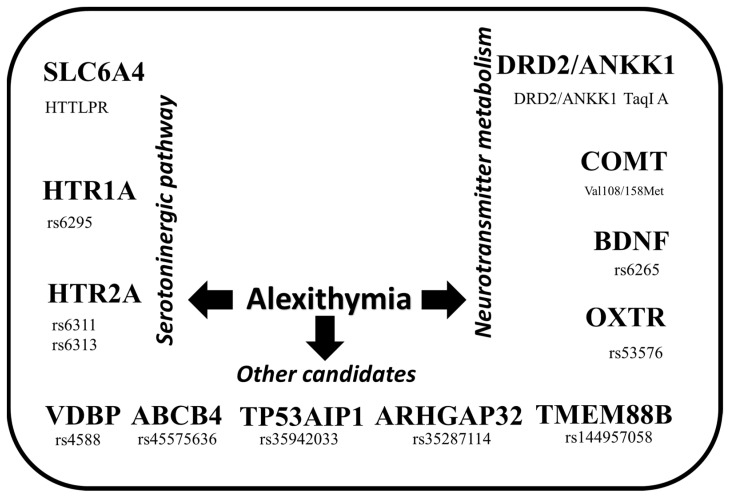
Main candidate genes associated with alexithymia.

**Table 1 genes-15-01025-t001:** Summary of genetic association studies on alexithymia.

First Author	Country	Sample Size	Gene	Cytogenetic Band	Variant	General Outcome	Gender Outcome	NOS Score
Kano M., 2012 [28]	Japan	304	*SLC6A4*	17q11.2	HTTLPR	Higher TAS-20 score with L/L genotype (*p* = 0.017).	Higher DIF domain score with L/L genotype (*p* = 0.001) in females.	7
Li X., 2020 [29]	China	1302	*HTR2A*	13q14.2	rs6311, rs6313	Higher TAS-20 score with G allele of rs6311 polymorphism (*p* = 0.014).	Higher TAS-20 score in males; after including gender as a covariate, rs6311 genotypes (*p* = 0.026) remained significantly different.	8
Mezzavilla M., 2015 [27]	Italy	585	*ABCB4, TP53AIP1, ARHGAP32, TMEM88B*	7q21.12, 11q24.3, 11q24.3, 1p36.33	rs45575636, rs35942033, rs35287114, rs144957058	Analysis of functional variants showed that these 4 polymorphisms are associated with alexithymia (*p* = 2.6 × 10^−8^, *p* = 3.2 × 10^−7^, *p* = 3.2 × 10^−7^, *p* = 1.0 × 10^−6^).	The average TAS-20 score in females was 46.1; in males, it was 45.3.	8
Terock J., 2021 [26]	Germany	6267	*HTR2A, HTR1A*	13q14.2, 5q12.3	rs6295, rs6313, rs6311	No association between polymorphisms in this study and alexithymia.	The average TAS-20 scores were 52.13 in males and 50.69 in females.	9
Koh MJ, 2016 [32]	Korean	244	*COMT*	22q11.21	Val158Met	Higher TAS-20 score with Val/Val genotype (*p* = 0.018).	No association.	8
Gong P., 2015 [32]	china	504	*HTR1A*	5q12.3	rs6295 (C-1019G)	Higher TAS-20 score with CG/GG genotype (*p* = 0.017) of rs6295.	Not reported.	6
Highland, K.B., 2011 [34]	USA	297	*DRD2/ANKK1*	11q23.2	TaqI A	Higher TAS-20 score with A1+ allele (*p* < 0.05).	Not reported.	7
Terock J., 2018 [25]	Germany	5283	*SLC6A4*	17q11.2	5-HTTLPR	Lower TAS-20 scores with L-allele (*p* = 0.022) of 5-HTTLPR.	Not reported.	9
Sacchinelli E., 2018 [24]	Italy	115	*SLC6A4*	17q11.2	HTTLPR	No association between *SCL6A4* and TAS-20 total score.	Higher DIF domain scores with L/L genotype (*p* = 0.036) in males.	7
Terock J., 2021 [17]	Germany	5783	*VDBP (GC)*	4q13.3	rs4588, rs7041	No association between rs4588 and rs7041 and alexithymia.	Not reported.	9
Koh M.J., 2016 [30]	Korea	355	*OXTR*	3p25.3	rs237885, rs237887, rs2268490, rs4686301, rs2254298, rs13316193, rs53576, rs2268498	No association between polymorphisms in this study and alexithymia.	Not reported.	6
Walter N.T., 2011 [23]	Germany	664	*BDNF, DRD2/ANKK1*	11p14.1, 11p14.1	Val66Met, TaqI A	Higher TAS-20 score with 66Met of Val66Met and A1 allele of *DRD2/ANKK1* (*p* < 0.02).	Significant effect of gender on the TAS-20 score (*p* = 0.015), indicating slightly higher TAS-20 scores in men than in women.	7
Ikarashi H., 2021 [18]	Japan	80	*COMT*	22q11.21	Val158Met	Higher DIF domain scores with *COMT* met carriers (*p* = 0.001).	Not reported.	7
Schneider-Hassloff H., 2016 [22]	Germany	195	*OXTR*	3p25.3	rs53576	No association of rs5376 and alexithymia.	No significant sex-specific genotype-by-CAS interaction was observed	7
Voigt G., 2015 [21]	Germany	144	*BDNF, DRD2/ANKK1*	11p14.1, 11p14.1	Val66Met, TaqI A	No association of *COMT* Val66Met, or TaqI A and alexithymia.	Not reported.	6
Ham B.J., 2005 [33]	Korea	109	*COMT* *5-HTTLPR*	22q11.21	Val108/158Met HTTLPR	Carriers of Val/Val genotype (*p* = 0.019) of *COMT* were associated with alexithymia. For HTTLPR, no association was observed.	Not associated.	7
Porcelli P., 2015 [20]	Italy	130	*HTR1A, SLC6A4*	5q12.3, 17q11.2	rs6295, HTTLPR	Homozygosity for *HTR1A*-G and 5-HTTLPR long alleles was associated with significantly (*p* < 0.01) higher TAS-20 scores.	Not reported.	8
	Sample size	22,361						

SLC6A4: solute carrier family 6 member 4, HTR1A: serotonin 1A receptor, HTR2A: serotonin 1A receptor; DRD2: dopamine receptor D2, ANKK1: ankyrin repeat and kinase domain containing 1, COMT: catechol-o-methyltransferase, BDNF: brain-derived neurotrophic factor, OXTR: oxytocin receptor, VDBP: vitamin D-binding protein, TP53AIP1: tumor protein P53 regulated apoptosis inducing protein 1, ARHGAP32: Rho GTPase Activating Protein 32, TMEM88B: Transmembrane Protein 88B, DIF: difficulty identifying feeling.

## Data Availability

No new data were created or analyzed in this study. Data sharing is not applicable to this article.

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
