# Peer review of "Exploring Candidate Gene Studies and Alexithymia: A Systematic Review"

_genes, 2024, doi:10.3390/genes15081025_

Round 1

Reviewer 1 Report

Comments and Suggestions for Authors
  • A brief summary

This manuscript summarizes the most recent literature on candidate gene studies about alexithymia. From 22,361 individuals, 12 genes and 13 polymorphisms were identified. Each gene's functional association with alexithymia was discussed.

  • General concept comments

1.     The result part of the abstract needs to be revised.

2.     Gender is also an exciting aspect of alexithymia. If the author can summarize the gender difference in the prevalence, this review will attract more readers. Then, in the results part for each candidate gene, any reported gender difference will be highlighted for this review.

  • Specific comments 

1.     In the introduction, line 45 states that 10% of the general population could be categorized as alexithymia, which comes from paper #1. As a review, a percentage range summarized from different studies will better describe the symptoms. Also, a gender effect should be mentioned.

2.     In the introduction, line 47 states that prevalence increases in adolescents and younger people. Another related study supports this statement and should be cited in this review 10.1371/journal.pone.0210519.

3.     Line 30 (PubMed, Embase, and Google Scholar) and line 73 (PubMed, Scopus, and Web of Science) conflict.

4.     Line 76, “performed a manual search,” specifies the search engine.

5.     Line 59, the genetic component, can usually be supported by a twin study. doi.org/10.1159/000056261 is an example.

6.     In Table 1, adding the p-value to each variant will be easy for readers to check.

7.     Like other candidate genes, Line 216's authors should include a more detailed summary of the gene function rather than only pointing out the association.

8.     Are lines 287-290 from the template?

Comments on the Quality of English Language

Moderate editing of English language required.

Author Response

REVIEW 1

GENERAL CONCEPT COMMENTS

Comment 1. The result part of the abstract needs to be revised.

Response. We deeply appreciate the reviewer´s suggestion, and have changed the abstract. We deleted part of methods section. And, we add augmented the results subsection in the results.

Change in the manuscript.

We used the words “Alexithymia” and “gene” and “genetics” and “variants” and “biomarkers”. The present systematic review was to perform following the Preferred Reporting Items for Systematic reviews and Meta-Analyses statement. We found only candidate gene studies. A total of seventeen studies met the eligibility criteria, which comprised 22,361 individuals. The candidate genes associated with alexithymia were, serotoninergic pathway SLC6A4, HTR1A, HTR2A, neurotransmitter metabolism, DRD2/ANKK1, COMT, BDNF, OXTR, other pathway, VDBP, TP53AIP1, ARHGAP32, and TMEM88B. (Page, 1, Lines 33-38)

Comment 2. Gender is also an exciting aspect of alexithymia. If the author can summarize the gender difference in the prevalence, this review will attract more readers. Then, in the results part for each candidate gene, any reported gender difference will be highlighted for this review.

Response. We deeply appreciate the reviewer’s 1 suggestion. According to the available data, we summarize some findings based on gender and we have inserted this in table 1.

Change in the manuscript.

Table 1, Gender outcome column. Page 5, Lines 145- Page 6 Lines 147.

Limitations. Another point to observe is the limitation of the data available regarding to the variable of gender. It has been stablished that gender could cause an imbalance because a combination of biological, psychological, and sociocultural factors that can be involved. Furthermore, given the distance between the specific protein encoded by a given gene (COMT, HTR1A, OXTR, among others) and the complexity of alexithymia that comprise a clinical diagnosis, it is more appropriate to identify discrete variables such a gender in further studies. (Page:10, Lines:326-332)

SPECIFIC COMMENTS

Comment 1. In the introduction, line 45 states that 10% of the general population could be categorized as alexithymia, which comes from paper #1. As a review, a percentage range summarized from different studies will better describe the symptoms. Also, a gender effect should be mentioned.

Response. Thanks for your comment we modified this section of the introduction.

Change in the manuscript.

Introduction section: Concerning to epidemiological data, it is difficult to estimate the prevalence of alexithymia due to there being no clear diagnostic criteria. In general population, around 10% of people having problematically high levels of alexithymia [1]. Nevertheless, in patients with physical illnesses alexithymia could reach more than 60% [2]. Moreover, gender differences have been pointed out; for example, large-scale studies in general population samples reported that males showed higher scores than females [3,4]. In fact, it has been reported that the prevalence of alexithymia ranged to 9-17% for men and 5-10% for women [5].. (Page:1-2, Lines:47-57)

Comment 2. In the introduction, line 47 states that prevalence increases in adolescents and younger people. Another related study supports this statement and should be cited in this review 10.1371/journal.pone.0210519.

Response. We added this information.

Change in the manuscript.

Introduction section: Specifically, in females, pubertal maturation was associated with difficulties identifying and describing feelings [8]. Hence, developmental changes in alexithymia should be considered during the adolescence. (Page:2, Lines:55-57)

Comment 3. Line 30 (PubMed, Embase, and Google Scholar) and line 73 (PubMed, Scopus, and Web of Science) conflict.

Response. Thanks you are right

Change in the manuscript.

Abstract section: Electronic databases including PubMed, Scopus, and Web of Science were searched for the study purpose. (Page:1, Lines:30-31)

Comment 4. Line 76, “performed a manual search,” specifies the search engine.

Response. We clarified this point

Change in the manuscript.

Search strategy: References in identified articles and reviews were hand-searched for relevant studies, and surfaced one articles that was not captured by the search strategy. (Page 2, Lines 88-91)

Comment 5. Line 59, the genetic component, can usually be supported by a twin study. doi.org/10.1159/000056261 is an example.

Response. Thanks, this was added.

Change in the manuscript.

Introduction section: In this sense, a twin study indicated that familial influences contributed to alexithymia. Specifically, this study support that genetic factors are implicated to externally oriented thinking domain [19]. (Page:2, Lines:70-72)

Comment 6. In Table 1, adding the p-value to each variant will be easy for readers to check.

Response. According to the reviewer´s suggestion, we have added the p-value in the Table 1.

Change in the manuscript.

Please see Table 1, General outcome column.

Comment 7. Like other candidate genes, Line 216's authors should include a more detailed summary of the gene function rather than only pointing out the association.

Response. Thanks we edited according to your comment

Change in the manuscript.

Other genetic pathways: Another polymorphism studied with alexithymia are of the vitamin D binding-protein (VDBP) gene. Related to, low vitamin D serum levels may lead to lower expression rated of the rate-limiting enzyme in serotonin synthesis, which is linked with an increase of the presynaptic shortcoming of available serotonin. In a general population of 5,744 individuals a study genotyped the VDBP gene variant rs4588 and rs7041; nevertheless, they report a lack significant association with alexithymia. Additionally, other study analyzed exome functional variants in association with alexithymia. The exome association analysis detected 4 significant variants (ABCB4 gene: rs45575636, TP53AIP1 gene: rs35942033, ARHGAP32 gene: rs35287114 and TMEM88B gene: rs144957058) associated with alexithymia. There is not much information about these genes and their biological pathways on the brain. ABCB4/Abcb4 mRNA expression is low but detectable in human brain and rat choroid plexus; it is an extremely effective phosphatidylcholine transporter. TP53AIP1 is thought to play an important role in mediating p53-dependent apoptosis. Meanwhile, ARHGAP32 encodes a neuron-associated GTPase-activating protein and is involved in early brain development, including extension of axons and dendrites, and postnatal remodeling and fine-tuning of neural circuits. Little is known about TMEM88B gene or its protein, so further studies are necessary to understand functional characterization. (Page:8, Lines:233-250).

Comment 8. Are lines 287-290 from the template?

Response. This was deleted.

Reviewer 2 Report

Comments and Suggestions for Authors

In this paper, Hernandez-Diaz et al. examined current literature and summarized genes associated with alexithymia. Specifically, they found that genes of serotoninergic pathway and neurotransmitter metabolism are related to alexithymia. However, significant caveats mentioned below suggest that this paper does not merit a publication on this journal.

1.       As a review paper, the authors talked about how they search for literature in abstract (line 30-34) and in methods. This information should never be brought up in a review paper as this should be part of authors’ “behind the scenes effort”. Knowing this information does not help the audience to understand alexithymia. However, the authors spend large amounts of space to emphasize this procedure which does not make sense to this reviewer.

2.       In the results section, the authors listed several genes related to alexithymia but mostly introducing their genomic location, gene size, isoforms etc. Again, this information is not interesting to the audience. In fact, it is never mentioned how exactly these genes function to affect neural behavior (i.e. the underlying mechanisms).

3.       The current discussion section is not a “discussion” but rather a repeat of the results section. The authors should briefly summarize the published work and discuss the open questions. In addition, unrelated paragraphs should not be included in line 287-290.

4.       A review paper should not have methods, results and discussion sections in the first place. The current manuscript should not be considered as a review.

5.       Figure 2 is confusing. What does the grey circle mean? The authors should improve the way they convey this idea.

In summary, significant improvement is needed to discuss the mechanisms of these genes. The format of this paper needs to be standardized.

Author Response

REVIEW 2

Comments and Suggestions for Authors

In this paper, Hernandez-Diaz et al. examined current literature and summarized genes associated with alexithymia. Specifically, they found that genes of serotoninergic pathway and neurotransmitter metabolism are related to alexithymia. However, significant caveats mentioned below suggest that this paper does not merit a publication on this journal.

Comment 1. As a review paper, the authors talked about how they search for literature in abstract (line 30-34) and in methods. This information should never be brought up in a review paper as this should be part of authors’ “behind the scenes effort”. Knowing this information does not help the audience to understand alexithymia. However, the authors spend large amounts of space to emphasize this procedure which does not make sense to this reviewer.

Response. We agree with the reviewer, and in the revised manuscript we deleted information about search for literature. And, we added that the present systematic review was to perform following the “Preferred Reporting Items for Systematic reviews and Meta-Analyses” (PRISMA) statement. (PMID: 33782057 PMID: 33781348). Therefore, given the limited number of words given by the journal and following the PRISMA criteria, we have the following checklist.

Change in the manuscript.

Abstract section: The present systematic review was to perform following the Preferred Reporting Items for Systematic reviews and Meta-Analyses statement. (Page 1, Lines 33-34).

Design section: The guidelines of the Preferred Reporting Items for Systematic Review and Me-ta-Analysis (PRISMA) were followed for the methodology and served as template for this present review. (Page:2, Lines:81-83)

Supplementary materials (Table S2).

Table S2. PRISMA Abstract Checklist

Topic

No.

Item

Reported?

TITLE

Title

1

Identify the report as a systematic review.

Yes

BACKGROUND

Objectives

2

Provide an explicit statement of the main objective(s) or question(s) the review addresses.

Yes

METHODS

Eligibility criteria

3

Specify the inclusion and exclusion criteria for the review.

Yes

Information sources

4

Specify the information sources (e.g. databases, registers) used to identify studies and the date when each was last searched.

Yes

Risk of bias

5

Specify the methods used to assess risk of bias in the included studies.

Yes

Synthesis of results

6

Specify the methods used to present and synthesize results.

Yes

RESULTS

Included studies

7

Give the total number of included studies and participants and summarise relevant characteristics of studies.

Yes

Synthesis of results

8

Present results for main outcomes, preferably indicating the number of included studies and participants for each. If meta-analysis was done, report the summary estimate and confidence/credible interval. If comparing groups, indicate the direction of the effect (i.e. which group is favoured).

Yes

DISCUSSION

Limitations of evidence

9

Provide a brief summary of the limitations of the evidence included in the review (e.g. study risk of bias, inconsistency and imprecision).

Yes

Interpretation

10

Provide a general interpretation of the results and important implications.

Yes

OTHER

Funding

11

Specify the primary source of funding for the review.

No

Registration

12

Provide the register name and registration number.

No

 From: Page MJ, McKenzie JE, Bossuyt PM, Boutron I, Hoffmann TC, Mulrow CD, et al. The PRISMA 2020 statement: an updated guideline for reporting systematic reviews. MetaArXiv. 2020, September 14. DOI: 10.31222/osf.io/v7gm2. For more information, visit: www.prisma-statement.org

Comment 2. In the results section, the authors listed several genes related to alexithymia but mostly introducing their genomic location, gene size, isoforms etc. Again, this information is not interesting to the audience. In fact, it is never mentioned how exactly these genes function to affect neural behavior (i.e. the underlying mechanisms).

Response. We apologized for the fact that we did not clearly describe the mechanisms in the results section. However, we follow the criteria of PRISMA guidelines; therefore, we underlying the probable mechanisms in alexithymia in the discussion section.

Change in the manuscript.

Discussion section: (Page:9, Lines:273-316

Comment3. The current discussion section is not a “discussion” but rather a repeat of the results section. The authors should briefly summarize the published work and discuss the open questions. In addition, unrelated paragraphs should not be included in line 287-290.

Response. Thanks, we deleted this paragraph Lines 273-316 were delated. Furthermore, we improved the discussion section according to your suggestion.

Change in the manuscript.

Discussion section: Due to, if these polymorphisms proposed for the transporter (HTTLPR) or receptors (rs6295 and rs6311) genes could leads to changes in 5-HT concentrations in the synaptic cleft or amygdala; subsequently, appears these type of behavioral traits. This is the principal reason why serotonin pathway is important for drug development [55,56]. Furthermore, it has been reported a modulatory effect of the 5-HTTLPR polymorphism on the connectivity of the ventral attention network and an impaired response inhibition, which are implicated in stimulus-driven attentional control [57]. Another candidate gene from serotoninergic system are the serotonin receptors. Specially, rs6295 and rs6311polymorphisms from HTR1A and HTR2A genes respectively. Derived from these investigations, we could found that G allele carriers seemed to be more vulnerable to alexithymia [26]. Serotonin is responsible for regulating a wide variety of mood disorders. Mostly through the action of serotonin receptors (HTR1A and HTR2A) and serotonin transporter (SERT) [55]. Changes in the disruption to the serotoninergic system in strongly linked with mood disorders. Therefore, different contribution of 5HT receptors (HTR1A and HTR2A) on the brain, should be considered in alexithymia [56]. Another import pathway related with the behavior are those involved in the neurotransmitter metabolism, such DRD2/ANKK1, COMT, BDNF and oxytocin receptor. First, A1+ allele of DRD2/ANKK1 gene is statistical liked with alexithymia [21,23]. A1 allele affects DRD2 receptor density and increases l-DOPA activity; indeed, this allele is more associated with negative emotions or cognitive alterations [58]. The above because these receptors are expressed primarily in sub-cortical regions like the nucleus accumbens and caudate putamen where they are involved in the modulation of emotions; highlighting the theirs importance in alexithymia [59]. Concerning to COMT the Val108/158Met polymorphism the Val carries are the individuals with a higher risk to alexithymia [32,33]. The Val form of the COMT enzyme is more active and leads to lower levels of synaptic dopamine in the prefrontal cortex [60]. Specifically, COMT Val108/158Met variation is linked with a increased activation in prefrontal areas causing a reduced ability in the specific domain and an increased cortico-limbic activity allowing an emotional dysregulation; which point to an association between COMT genotype and susceptibility for affective disorders such as alexithymia [61]. Therefore, Val108/158Met variant is highly related in genetic association studies with the variant of ANKK1/DRD2 genes.

In the other hand, according to BDNF gene, the candidate variant was Val66Met, specifically 66Met allele in patients with alexithymia. In BDNF protein, the substitution of Methionine disrupted the activity-dependent secretion of BDNF at the synapse, subsequently affects hippocampal functions [62]. The above, allow to associated BDNF gene to a predisposition to a higher risk in specific personality traits, including, fatigue, frustration, worrying, pessimism, shyness, persistence among others, including alexithymia. Finally, for OXTR gene one candidate variant for alexithymia risk is rs53576 polymorphism. This polymorphism is related to empathic responses. Precisely, the G allele carriers show greater empathic response the AA homozygotes [63]. In this idea, the expression of the previously mentioned genetic biomarkers of neurotransmitters metabolism is related to specific brain areas such as amygdala or posterior cingulate cortex. Specifically, these brain areas are involved in some emotional process (e.g., emotional stimuli, emotional experience, self-referential thinking) [64,65].

Alexithymia has been implicated in the prognosis and development of diverse diseases. Nevertheless, health workers seem to rarely consider alexithymia during treatment of any disease. Therefore, we consider very important the understanding of the complex genetic network leading to onset and progression of alexithymia. Nevertheless, the findings presented we want to recognize some limitations. First, alexithymia has a limited number of genetic association studies in comparison with other psychiatric disease or traits like schizophrenia or depression. In this idea, we not observed GWAS studies in Alexithymia. However, this is an exploratory view of the genetic background of alexithymia candidate biomarkers. This open the door for need the GWAS, epigenetic and expression studies. Another point to observe is the limitation of the data available regarding to the variable of gender. It has been stablished that gender could cause an imbalance because a combination of biological, psychological, and sociocultural factors that can be involved. Furthermore, given the distance between the specific protein encoded by a given gene (COMT, HTR1A, OXTR, among others) and the complexity of alexithymia that comprise a clinical diagnosis, it is more appropriate to identify discrete variables such a gender in further studies.

. (Page:9, Lines:274 to page 10, line 333)

Comment 4. A review paper should not have methods, results and discussion sections in the first place. The current manuscript should not be considered as a review.

Response. Our aim was to perform a systematic review following the “Preferred Reporting Items for Systematic reviews and Meta-Analyses” (PRISMA) statement. (PMID: 33782057 PMID: 33781348). The main Checklist is the following (Table S3):

Table S3. PRISMA 2020 Main Checklist

Topic

No.

Item

Location where item is reported

TITLE

Title

1

Identify the report as a systematic review.

1

ABSTRACT

Abstract

2

See the PRISMA 2020 for Abstracts checklist

INTRODUCTION

Rationale

3

Describe the rationale for the review in the context of existing knowledge.

1-2

Objectives

4

Provide an explicit statement of the objective(s) or question(s) the review addresses.

2

METHODS

Eligibility criteria

5

Specify the inclusion and exclusion criteria for the review and how studies were grouped for the syntheses.

2

Information sources

6

Specify all databases, registers, websites, organisations, reference lists and other sources searched or consulted to identify studies. Specify the date when each source was last searched or consulted.

2

Search strategy

7

Present the full search strategies for all databases, registers and websites, including any filters and limits used.

2

Selection process

8

Specify the methods used to decide whether a study met the inclusion criteria of the review, including how many reviewers screened each record and each report retrieved, whether they worked independently, and if applicable, details of automation tools used in the process.

2

Data collection process

9

Specify the methods used to collect data from reports, including how many reviewers collected data from each report, whether they worked independently, any processes for obtaining or confirming data from study investigators, and if applicable, details of automation tools used in the process.

2

Data items

10a

List and define all outcomes for which data were sought. Specify whether all results that were compatible with each outcome domain in each study were sought (e.g. for all measures, time points, analyses), and if not, the methods used to decide which results to collect.

2

10b

List and define all other variables for which data were sought (e.g. participant and intervention characteristics, funding sources). Describe any assumptions made about any missing or unclear information.

2

Study risk of bias assessment

11

Specify the methods used to assess risk of bias in the included studies, including details of the tool(s) used, how many reviewers assessed each study and whether they worked independently, and if applicable, details of automation tools used in the process.

2-3

Effect measures

12

Specify for each outcome the effect measure(s) (e.g. risk ratio, mean difference) used in the synthesis or presentation of results.

3

Synthesis methods

13a

Describe the processes used to decide which studies were eligible for each synthesis (e.g. tabulating the study intervention characteristics and comparing against the planned groups for each synthesis (item 5)).

2-3

13b

Describe any methods required to prepare the data for presentation or synthesis, such as handling of missing summary statistics, or data conversions.

NA

13c

Describe any methods used to tabulate or visually display results of individual studies and syntheses.

Figure 1

13d

Describe any methods used to synthesize results and provide a rationale for the choice(s). If meta-analysis was performed, describe the model(s), method(s) to identify the presence and extent of statistical heterogeneity, and software package(s) used.

NA

13e

Describe any methods used to explore possible causes of heterogeneity among study results (e.g. subgroup analysis, meta-regression).

NA

13f

Describe any sensitivity analyses conducted to assess robustness of the synthesized results.

NA

Reporting bias assessment

14

Describe any methods used to assess risk of bias due to missing results in a synthesis (arising from reporting biases).

NA

Certainty assessment

15

Describe any methods used to assess certainty (or confidence) in the body of evidence for an outcome.

2

RESULTS

Study selection

16a

Describe the results of the search and selection process, from the number of records identified in the search to the number of studies included in the review, ideally using a flow diagram.

3

16b

Cite studies that might appear to meet the inclusion criteria, but which were excluded, and explain why they were excluded.

3

Study characteristics

17

Cite each included study and present its characteristics.

3

Risk of bias in studies

18

Present assessments of risk of bias for each included study.

NA

Results of individual studies

19

For all outcomes, present, for each study: (a) summary statistics for each group (where appropriate) and (b) an effect estimate and its precision (e.g. confidence/credible interval), ideally using structured tables or plots.

Table 1

Results of syntheses

20a

For each synthesis, briefly summarise the characteristics and risk of bias among contributing studies.

Table 1

20b

Present results of all statistical syntheses conducted. If meta-analysis was done, present for each the summary estimate and its precision (e.g. confidence/credible interval) and measures of statistical heterogeneity. If comparing groups, describe the direction of the effect.

NA

20c

Present results of all investigations of possible causes of heterogeneity among study results.

NA

20d

Present results of all sensitivity analyses conducted to assess the robustness of the synthesized results.

NA

Reporting biases

21

Present assessments of risk of bias due to missing results (arising from reporting biases) for each synthesis assessed.

NA

Certainty of evidence

22

Present assessments of certainty (or confidence) in the body of evidence for each outcome assessed.

Table 1

DISCUSSION

Discussion

23a

Provide a general interpretation of the results in the context of other evidence.

7-9

23b

Discuss any limitations of the evidence included in the review.

9

23c

Discuss any limitations of the review processes used.

9

23d

Discuss implications of the results for practice, policy, and future research.

9

OTHER INFORMATION

Registration and protocol

24a

Provide registration information for the review, including register name and registration number, or state that the review was not registered.

NA

24b

Indicate where the review protocol can be accessed, or state that a protocol was not prepared.

NA

24c

Describe and explain any amendments to information provided at registration or in the protocol.

NA

Support

25

Describe sources of financial or non-financial support for the review, and the role of the funders or sponsors in the review.

9

Competing interests

26

Declare any competing interests of review authors.

9

Availability of data, code and other materials

27

Report which of the following are publicly available and where they can be found: template data collection forms; data extracted from included studies; data used for all analyses; analytic code; any other materials used in the review.

NA

From: Page MJ, McKenzie JE, Bossuyt PM, Boutron I, Hoffmann TC, Mulrow CD, et al. The PRISMA 2020 statement: an updated guideline for reporting systematic reviews. MetaArXiv. 2020, September 14. DOI: 10.31222/osf.io/v7gm2. For more information, visit: www.prisma-statement.org

Comment 5. Figure 2 is confusing. What does the grey circle mean? The authors should improve the way they convey this idea.

Response. We modified the figure 2 in order to be easier to understand.

In summary, significant improvement is needed to discuss the mechanisms of these genes. The format of this paper needs to be standardized.

Response. Thanks this was improved

Round 2

Reviewer 2 Report

Comments and Suggestions for Authors

The authors addressed all my concerns about their initial submission.